# Multi-scale Neural Networks for Retinal Blood Vessels Segmentation

## Abstract

Existing supervised approaches didn't make use of the low-level features which are actually effective to semantic segmentation. And another deficiency is that they didn't consider the relation between pixels, which means effective features are not extracted. In this paper, we applied semantic segmentation on retinal blood vessel images and proposed a novel convolutional neural network which make sufficient use of low-level features together with high-level features and involves atrous convolution to get multi-scale features which should be considered as effective features. Our model is tested on three standard benchmarks - DRIVE, STARE, and CHASE databases. The results presents that our model significantly outperforms existing approaches in terms of accuracy, sensitivity, specificity, the area under the ROC curve and the highest prediction speed. Our work provides evidence of the power of wide and deep neural networks in retinal blood vessels segmentation task which could be applied on other medical images tasks.

## 1 Introduction

Over the past years, retinal blood vessels have found a wide range of applications in medicine and health [1, 2]. As the only deep vessels that can be observed in human body, retinal blood vessels could directly reflect the omen of some cardiovascular diseases and could also reflect the severity of diabetic retinopathy [3]. With these properties, retinal blood vessels play a significant role in modern medicine.

The most essential step in diagnosis on retinal blood vessel is to identify the vessels from retinal images. Compared to manual diagnosis, automatic machine diagnosis in retinal images can reduce the probability of medical misdiagnosis. Machine retinal vessels diagnosis, known as the segmenatation of retinal blood vessels, is typically under the semantic segmentation task and has been studied with both unsupervised and supervised machine learning models. In conventional unsupervised approaches, features are manually extracted from images and are fed into a statistical model to detect blood vessels with on a threshold. Ricci et al. proposed a line detector to detect blood vessels at each pixel [4]. Unsupervised approaches doesn't require labeled data, which can be considered as an advantage. However, the performance of such approaches is far below the satisfaction in real application scenarios and most of them are time consuming. Supervised segmentation approaches then have been proposed to overcome these limitations. The main idea of these supervised models is to discriminate vessel pixels against non-vessel ones with a binary classifier which is trained with features extracted from annotated images. Recent studies of deep neural networks which could extract features automatically leads to the boom of supervised deep learning approaches in the segmentation of retinal blood vessels. Ronneberger et al. proposed U-net, which first uses Convolutional Neural Network (CNN) to extract the features from the original medical image and then upsamples these features to a segmentation image [8].

These deep learning based supervised approaches achieve much higher performance, compared to conventional supervised and unsupervised models, but still have some weaknesses which cause the bottleneck of performance. One of these weaknesses is the insufficient use of low-level features

1st Conference on Medical Imaging with Deep Learning (MIDL 2018), Amsterdam, The Netherlands.

which is helpful in semantic segmentation. Another weakness is that features extracted by existing approaches are not effective. To solve the first issue, fully convolutional neural network is involved so that low-level features can be obtained. To solve the second issue, multi-scale information should be extracted from image. Motivated by downsampling process in which multi-scale information can be obtained with different scales, we involved atrous convolution [41] in our convolutional neural network. In conclusion, We proposed a novel patch-based deep convolutional neural network which makes use of both high-level features and low-level features with context information, to segment retinal blood vessels.

The mainly parts of our method are as follows. First, an image is pre-processed by using improved automatic color enhancement, global contrast normalization and augmentation. Then, a patch-based network, which is able to reduce the unfavorable phenomenon of overfitting, is used as the classifier to perform supervised detection of vessels. Particularly, in order to obtain the context information such as location and edge, the proposed network makes use of cross-layer connections to incorporate low-level features into the densely connected neural network. We evaluated the proposed method using three publicly available datasets as DRIVE, STARE and CHASE. Results show the superior performance of our method over several existing methods in terms of four criteria, sensitivity, specificity, classification accuracy and the area under the ROC curve.

## 2    Related work

Retinal blood vessel segmentation belongs to semantic segmentation. In general, segmentation methods can be divided into unsupervised and supervised. In unsupervised approaches, features are extracted manually and then feed to a statistical learning model which doesn't required labeled data. Ricci et al. uses line detectors to detect blood vessels at each pixel [4]. Villalobos-Castaldi et al. first obtained features from the co-occurrence matrix of retinal images and then filter out some features by a threshold [5]. Azzopardi et al. detected vessels by combining co-linearly aligned Gaussian filters under the consideration of actual patterns such as straight vessel and crossover points customized by a user [6]. Zhao et al. adopted a sophisticated active contour model that involves pixel brightness and features extraction [7]. In supervised approaches, the necessary information are extracted from images with annotated data. The basic idea behind supervised models is using extracted features to train a classifier to discriminate between vessel and non-vessel pixels. In most cases, supervised models achieve better performance and save lots of time, compared to unsupervised approaches. Fraz et al. used gradient vector field, morphological transformation, line feature and Gabor responses to get features and trained a neural network [9]. Marin et al. used gray-level and 7-D feature vectors to train a neural network [10].

In recent years, CNN has achieved great success in the field of visual perception, and the semantic segmentation of images is one of the successful cases. Qiaoliang et al. adopted the full convolution network based on patches and a sliding window to train a five-layer neural network. The reported accuracy, sensitivity, specificity and the area under ROC curve of their method are 0.9527, 0.7569, 0.9816 and 0.9738, respectively, on the DRIVE database and 0.9628, 0.7726, 0.9844 and 0.9879 for the STARE database and the average time for processing one image is 1.2 min [11]. Liskowski et al. used global contrast normalization, zero-phase whitening, and augmented using geometric transformations and gamma corrections to get feature vectors to train their deep neural network and they also used a structure prediction architecture to get better results [12]. The average accuracy, sensitivity, specificity and the area under ROC curve are improved to 0.9535, 0.7811, 0.9807 and 0.9790 for DRIVE and 0.9729, 0.8554, 0.9862 and 0.9928 for STARE and the average time for processing one image is 1.5 min. In our work, the method proposed in this paper is also classified as supervised and to our best knowledge, the proposed method outperforms all state-of-art results.

## 3    Models

In this section, we will first introduce our overall model, which adopts fully convolutional neural network architecture. And then two novel sub architectures will be introduced to solve the bottleneck most existing models have. And we suppose the retinal images fed to the network have been preprocessed, which will be mentioned in section 4.

### 3.1 Overall architecture

The overall architecture is shown in the left-most part in Figure 1. Our model adopts the fully convolutional neural network which is commonly used in most semantic segmentation tasks. It can be briefly divided into two parts - encoder and decoder. The encoder is a convolutional neural network which extracts features from the input image such as the retinal blood vessels image and the decoder will upsample the extracted features to the result image that we desired, such as the vessels segmentation in our task. In the following sections, we will introduce two novel architectures that used in our fully convolutional neural network.

### 3.2 Convolution-Only Neural Network

Pooling layer will cause the missing of some information. Therefore we choose to use convolution operation with stride 2 to replace pooling operation so that the information could be preserved as much as possible and the result feature map has the same size as in pooling layer. This architecture is shown in Figure 1 (1).

Another modification in the convolutional layer is that we fuse the features extracted from two adjacent convolutional layers together. So that the low-level information could be passed to the top layers and the model could make sufficient use of the effective low-level features to do semantic segmentation. This architecture is shown in Figure 2.

### 3.3 Multi-scale sensitive Atrous Convolution

**Atrous Convolution** Atrous convolution proposed by Fisher et.al [22] is an operation that adds "holes" in normal convolutional operation so that the resolution of features could be controlled by these "holes". Another property of atrous convolution is that it could reduce the parameters in network but expand the reception fields. To clarify atrous convolution operation, we borrow the mathematical descriptions from Fisher et.al [22]:

Let $F : \mathbb{Z}^2 \to \mathbb{R}$ be a discrete function. Let $\Omega_l = [-l, l]^2 \cap \mathbb{Z}^2$, let $k : \Omega_l \to \mathbb{R}$ be a discrete filter of size $(2l + 1)^2$ and let $r$ be a atrous convolution rate and let $*_r$ be a atrous convolution operation, defined as

$$(F *_r k)(p) = \sum_{(s+rt)=p} F(s)k(t) \tag{1}$$

And the normal convolution operation is the atrous convolution with rate equals to one. The details of atrous convolution can be found in [22].

**Multi-scale sensitive features** In order to get multi-scale information, we apply 6 atrous convolutional layers with different rate, one max pooling layer and one average pooling layer simultaneously on the input image, and then concatenate the extracted feature maps to obtain the effective multi-scale feature map, as shown in Figure 1 (2).

## 4 Experiments

### 4.1 Data sources

We used three publicly available databases - DRIVE, STARE and CHASE to evaluate our model.

The DRIVE dataset was obtained from a diabetic retinopathy screening program in the Netherlands which consisted of 400 diabetic subjects between 25 - 90 years of age, but only 40 images were randomly selected for training and testing which both contain 20 images. For the 40 images, there are 7 images show signs of mild early diabetic retinopathy. In addition, all the images were made by 3CCD camera and each has size of $565 \times 584$. For each image, a corresponding mask image is also provided. (`http://www.isi.uu.nl/Research/Databases/DRIVE/`)

The STARE dataset consists of 20 retinal fundus slides captured by a TopCon TRV-50 fundus camera. All the images have the size of $700 \times 605$. Half of the dataset comprises images of healthy subjects,

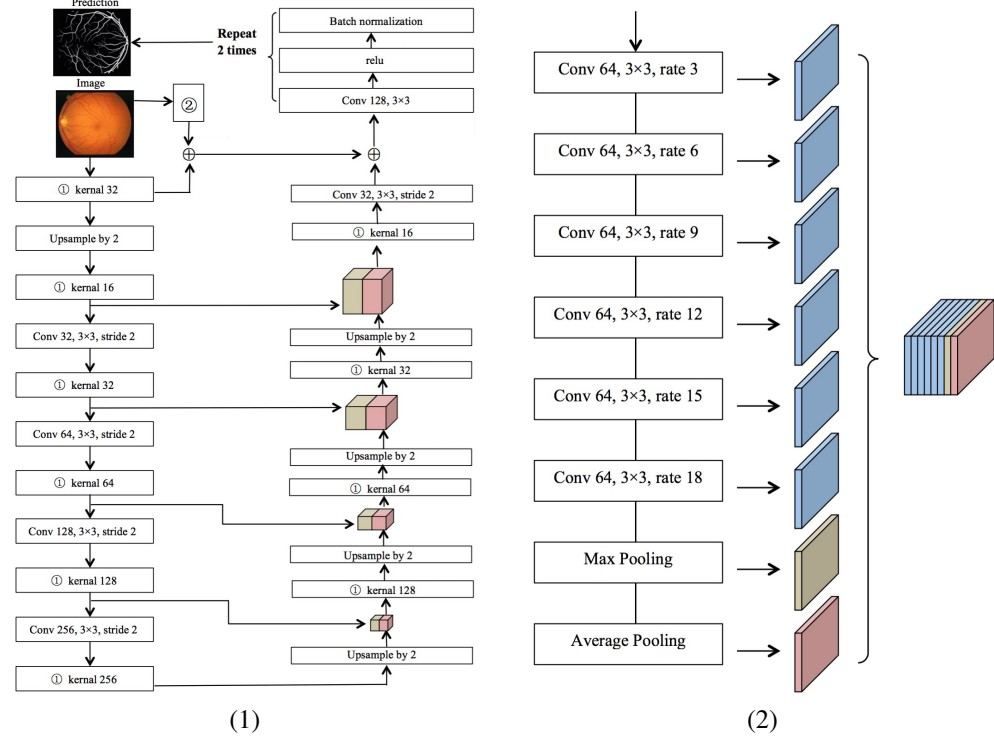

Figure 1: (1) is the overall model, (2) is the multi-scale architecture ②

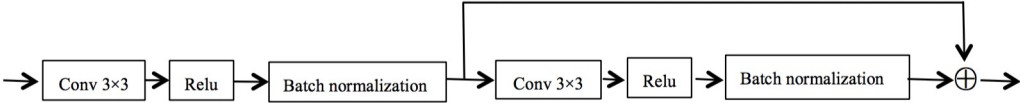

Figure 2: The two convolutional architecture ①

and the rest contains the pathological cases. As widely known, the pathological example makes the segmentation more challenging. (`http://www.ces.clemson.edu/~ahoover/stare/`)

The CHASE dataset is a subset of retinal images of multiethnic children from the Child Heart and Health Study in England and comprises 28 images with a resolution of $1280 \times 960$ pixels. (`https://blogs.kingston.ac.uk/retinal/chasedb1/`)

## 4.2 Image preprocessing

Neural networks can perform better on appropriately preprocessed images and we describe our preprocessing as follows.

**Global Contrast Normalization (GCN)** Think of a brain neuron which is only connected to a spatially global amount of visual receptor neurons, it seems to better respond to spatial directions rather than precise locations. For the purpose of better response of neural networks, we perform global contrast normalization that every patch of retinal images is standardized by subtracting the mean and dividing by the standard deviation of its elements.

**Automatic Color Enhancement (ACE)** Besides GCN, the automatic color enhancement (ACE), which is a smoothed and localized modification method of the normalized histogram equalization, adjusts the contrast of retinal images to achieve the perceptual constancy of color and brightness. For the purpose of correcting the final value of pixel in each retinal image, the ACE mainly calculates the relationship with light and dark by the difference method and achieves a good enhancement effect. The outcome of ACE used for one retinal image is shown in Figure 3.

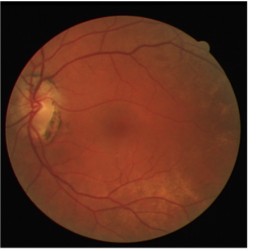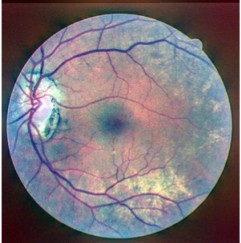

Figure 3: The outcome of ACE for one image.

The ACE is divided into three steps. First, we separate the RGB channels of one retinal image $I(x)$ with domain $\Omega$ and scale intensity values to $[0, 1]$. Then, we calculate the $R(x)$ value of each pixel to adapt local image contrast:

$$R(x) = \sum_{y \in \Omega/\ x} \frac{s_a(I(x) - I(y))}{\|x - y\|} \quad x \in \Omega \tag{2}$$

where $\Omega/\ x$ denotes $\{y \in \Omega : y \neq x\}$, $\|x - y\|$ denotes Euclidean distance, and $s_a : [-1, 1] \to R$ is the slope function $s_a(t) = \min\{\max\{\alpha t, -1\}, 1\}$ for some $\alpha \geq 1$. For the purpose of the global white balance, we extend $R(x)$ to $[0, 1]$. Finally, we solve the optimization problem. For one retinal image, the complexity of the ACE is $O(N^4)$ and it is too slow to handle all retinal images. Therefore, we use an improved ACE algorithm proposed by Pascal Getreuer [13]. The new algorithm adopts two approximate methods that they both significantly improve the speed of ACE. One uses polynomial function to approximate the slope function to reduce the calculation and the other uses different degrees of interpolation.

**Augmentations** Although GCN and ACE achieve a good enhancement effect, they cannot prevent overfitting since the number of images in DRIVE, STARE and CHASE datasets is small. To overcome this limitation, we use augmentations to increase the number of retinal images to make our model have a good generalization and prevent overfitting.

First, centering on each pixel, crop a $64 \times 64$ patch. And we can get a $64 \times 64 \times 3$ vector of each patch for retinal images, since images have three channels. It should be pointed out that extracting patches after preprocess methods is under the consideration of training and test speed. Then, to further explore augmentations, each patch of retinal images is composed of limited four randomized actions as follows: a) Flipping horizontally or vertically; b) Gamma correction of Saturation and Value (of the HSV colorspace) by raising pixels to a power in $[0.25, 4]$; c) Rotation by an angle in $[-90, 90]$; d) Scaling by a factor between $0.7$ and $1.3$.

### 4.3 Network configuration

**Training parameters** The training of network is carried out by adaptive moment estimation that is equivalent to the RMSprop with momentum items [14, 15]. The adaptive moment estimation adjusts the learning rate of each parameter dynamically by using two-moment estimation of the gradient. In addition, the iterative learning rate of adaptive moment estimation, which is in a certain range, makes the parameters more stable. The learning rate is initially set to $0.001$ and the training of network stops after $12,000$ iterations (9 epochs). The implementation, which is based on the Tensorflow framework [21], performs all computation on GPUs in single-precision arithmetic. Briefly, the experiments are conducted on the hardware configurations: Intel CPU E5-2680 with one NVIDIA GTX 1080Ti graphics card.

**Structure prediction** Not only the training of our network, but also the different structure predictions play an important role in the accuracy of segmentation. Furthermore, the different structure predictions improve test speed greatly. The structure prediction is mainly adopted in the testing by using the correlation between each pixel with its surrounding pixels of the retinal images. Like the patch-based average prediction method, different sliding window have different size and stride. Particularly, the edge pixels of the retinal images are predicted with less time due to the limitation.The windows of

sp-1, sp-32 and sp-64 are the whole image, $32 \times 32$ and $64 \times 64$ respectively. The stride of sp-1, sp-32 and sp-64 are 1, 3 and 3 respectively.

**Evaluation metrics** The evaluation metrics of our experiment is divided into two parts, one is in terms of area under ROC curve AUC, accuracy Acc, sensitivity Sens, specificity Spec and the other is the test speed that process one image (average time).

# 5 Results

## 5.1 Performance on the DRIVE dataset

We evaluated the performance of the proposed method on the DRIVE dataset and the results are shown in table 1 in Appendix. The results present that our model outperforms several existing methods. In AUC, our method achieves 0.9826, which is 2.12% higher than the state-of-the-art unsupervised method proposed by Azzopardi et al.and 0.36% higher than the state-of-the-art supervised method proposed by Liskowski et al. In classification accuracy, our method leads unsupervised methods by at least 1.39% and supervised methods by at least 0.46%. In sensitivity, our method leads the unsupervised methods by at least 3.78% and supervised methods by at least 2.22%. In specificity, our method leads unsupervised methods by at least 1.04% and supervised methods by at least 0.01%. As compared with ground truth (left) in Fig. 4 (a and b), our segmentation results (right) are smoother andvery close to the ground truth. And the best model of ours is SP-64, which have the highest AUC, accuracy and sensitivity. In summary, our model leads the AUC, accuracy and specificity of baseline by at least 0.6200%, 0.7300% and 1.200% respectively.

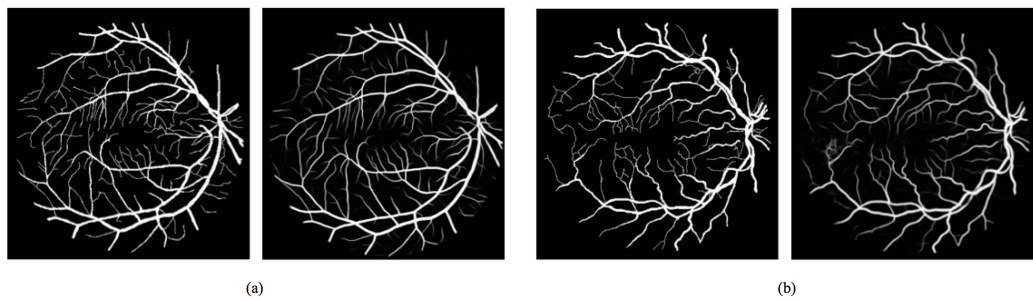

(a)                 (b)

Figure 4: Ground truth (left) and segmentation result (right) for one healthy subject: (a) DRIVE image #13, and one subject with pathologies: (b) DRIVE image #8.

## 5.2 Performance on the STARE dataset

The performance on STARE dataset is shown in table 2 in Appendix. From the results, our model have better performance then several existing methods. Particularly, In AUC, our model achieves 0.9930, which is 4.33% higher than Azzopardi's unsupervised approach and 0.02% higher than Liskowski's supervised method. In classification accuracy, our method leads unsupervised methods by at least 1.69% and supervised methods by at least 0.03%. In sensitivity, our method leads the unsupervised methods by at least 8.63% and supervised methods by at least 0.25%. In specificity, our method leads unsupervised methods by at least 1.61% and supervised methods by at least 0.0027%. These results are also confirmed by the comparison of ground truth(left) and segmentation result(right) in Fig. 5(a) and Fig. 5(b). In summary, our method also achieves better performance than several existing method for the same criteria although the improvement of the STARE dataset is lower than that on the DRIVE dataset and this may be due to the high AUC that exceeds 99%.

## 5.3 Performance on the CHASE dataset

The performance on CHASE dataset is shown in table 3 in Appendix. Our model outperforms Azzopardi's unsupervised approach in all criteria. Compared to existing supervised approaches, our model leads the best performance. In AUC, we achieves 98.65%, which is higher than the best existing model proposed by Fraz et al who achieved 97.60%. In classification accuracy, our model

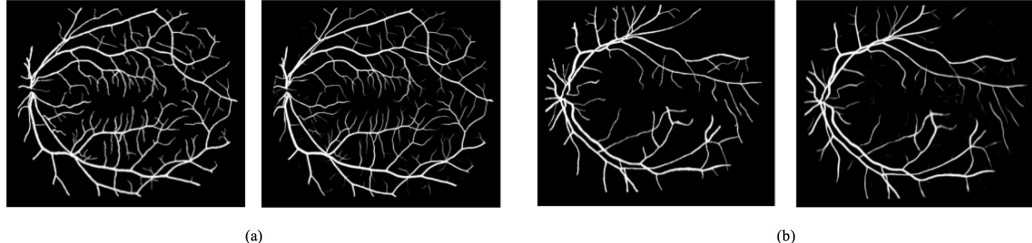

(a)           (b)

Figure 5: Ground truth (left) and segmentation result (right) for one healthy subject: (a) STARE image #0255, and one subject with pathologies: (b) STARE image #0002.

leads Li's approach by at least 0.81%. In specificity, our model leads Li's approach by at least 2.35%. In specificity, our model leads Li's approach by at least 0.83%. In summary, our method also achieves better performance than several existing method in CHASE dataset.

### 5.4  Test speed

Considering that the test speed is important to applications of retinal blood vessels besides above criteria, we evaluate the test time of our model. The results of our model exceed several existing methods that it needs less training time and has the highest test speed. it shows the inspiring test time in TABLE 4. The fastest version of our model only needs 3 seconds for each image, which has greatly exceeded several existing methods. In addition, we can see SP-1, SP-32 and SP-64 have different test speeds in TABLE 4, but the results in TABLE 3 show that SP-1, SP-32 and SP-64 have nearly the same performance. These results of SP-1, SP-32 and SP-64 lead us to conclude that our model speeds up testing while ensuring high accuracy. In summary, the SP-1 achieves the test time of 3s, which is 7s faster than the state-of-the-art unsupervised method [6], and 87s faster than the state-of-the-art supervised method [12].

### 5.5  Robustness

Although our method has the high accuracy and fast test speed, robustness also should be evaluated. We perform additional experiments on the above databases. Furthermore, we also consider three same problems according to Li et al. (Figure. 6 (2)) as follows [11].

**Segmentation in the presence of lesions** The retinal image of Fig. 6 (1) (a), which is from the STARE dataset, contains light and dark lesions of diabetic retinopathy. Compared with the real blood vessels, the probability value produced by our method is significantly lower for both bright and dark lesions and this shows the segmentation is less influenced by the presence of lesions.

**Segmentation in the presence of central vessel reflex** We choose one retinal image from the CHASE dataset (the Image_13L in Fig. 6 (1) (b)). The micro vessels, vessels without central reflex, vessels with central reflex in region 1 and vessels with central reflex in region 2 are correctly recognized by the proposed method.

**Segmentation of micro vessels with low contrast** In Fig. 6 (1) (c) (from DRIVE database), the 13_test contains micro vessels surrounding the macular and the 19_test contains micro vessels with very low contrast. The macula has low image intensity and it is close to the intensity of vessels in the green channel. With the above considerations, it is important to recognize the difference between it and micro vessels with very low contrast. Meanwhile, the segmentation indicates that the micro vessels surrounding the macular are correctly classified, the entire macular is recognized as a non-vessel and the micro vessels with very low contrast are also classified with high accuracy.

## 6  Conclusions

In this paper, we propose a novel neural network which make use of row-level features and could obtain multi-scale information to do vessel segmentation. We evaluate our network on three datasets - DRIVE, STARE and CHASE and get state-of-art results.

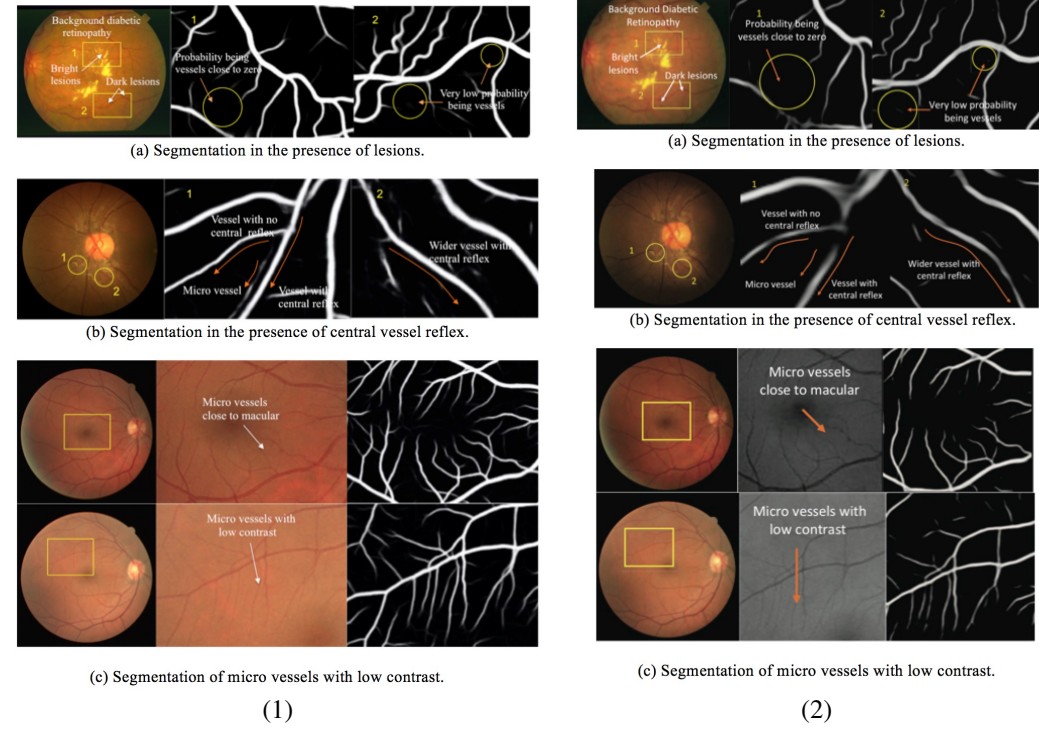

Figure 6: (1) and (2) are the segmentation in the presence of lesions, central vessel reflex and micro vessels of our methods and Li et al. [11] respectively. (a) Segmentation in the presence of lesions. (b) Segmentation in the presence of central vessel reflex. (c) Segmentation of micro vessels with low contras.

Further improvements could be achieved by a more thorough survey of network architectures, training parameters, structure prediction and preprocessing methods. All these experiment results confirm the superior performance of proposed deep neural networks as a vessel segmentation technique for fundus imaging.

### Acknowledgments

The authors declare that they have no competing interests.

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

# A   Appendix

Table 1: Performance of the proposed method on the DRIVE database

| No | Type | Methods | Year | AUC | Acc | Sens | Spec |
|----|------|---------|------|-----|-----|------|------|
| 1 | | Al-Diri [16] | 2009 | N.A | N.A | 0.7272 | 0.9551 |
| 2 | Unsupervised methods | Fraz [3] | 2011 | N.A | 0.9430 | 0.7152 | 0.9759 |
| 3 | | You [17] | 2011 | N.A | 0.9434 | 0.7410 | 0.9751 |
| 4 | | Azzopardi [6] | 2015 | 0.9614 | 0.9442 | 0.7655 | 0.9704 |
| 5 | | Niemeijer [18] | 2004 | 0.9294 | 0.9416 | N.A | N.A |
| 6 | | Ricci [4] | 2007 | 0.9558 | 0.9595 | N.A | N.A |
| 7 | | Lupascu [19] | 2010 | 0.9561 | 0.9597 | 0.7200 | N.A |
| 8 | | Marin [10] | 2011 | 0.9558 | 0.9452 | 0.7067 | 0.9801 |
| 9 | Supervised methods | Fraz [9] | 2012 | 0.9747 | 0.9480 | 0.7406 | 0.9807 |
| 10 | | Li [11] | 2015 | 0.9738 | 0.9527 | 0.7569 | 0.9816 |
| 11 | | Liskowski [12] | 2016 | 0.9790 | 0.9535 | 0.7811 | 0.9807 |
| 12 | | unet | 2018 | 0.9791 | 0.9536 | 0.7810 | 0.9807 |
| 13 | | SP-1 (ours) | 2018 | 0.9820 | 0.9577 | 0.7973 | 0.9811 |
| 14 | | SP-32 (ours) | 2018 | 0.9822 | 0.9579 | 0.8004 | 0.9809 |
| **15** | | **SP-64 (ours)** | **2018** | **0.9826** | **0.9581** | **0.8033** | **0.9808** |

Table 2: The comparison with other methods in last few years of STARE database

| No | Type | Methods | Year | AUC | Acc | Sens | Spec |
|----|------|---------|------|-----|-----|------|------|
| 1 | | Al-Diri [16] | 2009 | N.A | N.A | 0.7521 | 0.9681 |
| 2 | Unsupervised methods | Fraz [3] | 2011 | N.A | 0.9442 | 0.7311 | 0.9680 |
| 3 | | You [17] | 2011 | N.A | 0.9497 | 0.7260 | 0.9756 |
| 4 | | Azzopardi [6] | 2015 | 0.9497 | 0.9563 | 0.7716 | 0.9701 |
| 5 | | Ricci [4] | 2007 | 0.9602 | 0.9584 | N.A | N.A |
| 6 | | Fraz [9] | 2012 | 0.9768 | 0.9534 | 0.7548 | 0.9763 |
| 7 | | Li [11] | 2015 | 0.9879 | 0.9628 | 0.7726 | 0.9844 |
| 8 | Supervised methods | Liskowski [12] | 2016 | 0.9928 | 0.9729 | 0.8554 | 0.9862 |
| 9 | | unet | 2018 | 0.9850 | 0.9612 | 0.7887 | 0.9815 |
| **10** | | **SP-1 (ours)** | **2018** | **0.9930** | **0.9732** | **0.8579** | **0.9862** |
| 11 | | SP-32 (ours) | 2018 | 0.9923 | 0.9727 | 0.8544 | 0.9857 |
| 12 | | SP-64 (ours) | 2018 | 0.9926 | 0.9731 | 0.8520 | 0.9864 |

Table 3: The comparison with other methods in last few years of CHASE database

| No | Type | Methods | AUC | Acc | Sens | Spec |
|----|------|---------|-----|-----|------|------|
| 1 | Unsupervised methods | Azzopardi [6] | 0.9487 | 0.9387 | 0.7585 | 0.9587 |
| 2 | | Fraz [9] | 0.9712 | 0.9469 | 0.7224 | 0.9711 |
| 3 | | Li [11] | 0.9716 | 0.9581 | 0.7507 | 0.9793 |
| 4 | Supervised methods | Fraz [20] | 0.9760 | 0.9524 | 0.7259 | 0.9770 |
| 5 | | unet | 0.9809 | 0.9592 | 0.7113 | 0.9842 |
| **6** | | **SP-1 (ours)** | **0.9865** | **0.9662** | **0.7742** | **0.9876** |
| 7 | | SP-32 (ours) | 0.9861 | 0.9657 | 0.7736 | 0.9871 |
| 8 | | SP-64 (ours) | 0.9862 | 0.9657 | 0.7742 | 0.9872 |

Table 4: The comparison of average test time for one image with other methods in last few years

| Type | Method | Year | Processing |
|------|--------|------|-----------|
| Unsupervised methods | Al-Diri [16] | 2009 | 11 min |
| | Azzopardi [6] | 2015 | 10 s |
| Supervised methods | Marin [10] | 2011 | 1.5 min |
| | Fraz [9] | 2012 | 2 min |
| | Li [11] | 2015 | 1.2 min |
| | Liskowski [12] | 2016 | 1.5 min |
| | **SP-1** | **2018** | **3 s** |
| | SP-32 | 2018 | 1.3 min |
| | SP-64 | 2018 | 5 min |

