# OpenReview forum: "Multi-scale Neural Networks for Retinal Blood Vessels Segmentation"
_MIDL.amsterdam/2018/Conference — Submitted to MIDL 2018_

### Review · AnonReviewer2 · 2018-05-04
**Review of Multi-scale Neural Networks for Retinal Blood Vessels Segmentation**

**Rating:** 1
**Confidence:** 2

**Review:**

This study demonstrates a method for automated segmentation of blood vessels in retinal images. This study shows impressive segmentation results but has a number of flaws.

Comments:

1. There are a number of unsupported claims in the abstract and introduction, such as:

"Existing supervised approaches didn’t make use of the low-level features which
are actually effective to semantic segmentation."

"Compared to manual diagnosis, automatic machine diagnosis in retinal images can reduce the probability of medical misdiagnosis"

"However, the performance of such approaches is far below the satisfaction in real application scenarios and most of them are time consuming."

"Another weakness is that features extracted by existing approaches are not effective."

Adding supporting literature and evidence to these claims would allow for better assessment of this paper's novelty and importance.

2. There is a citation of non-existent reference [41].

3. The dataset description is unclear. Why were only 40 images selected from the DRIVE dataset? The images from different datasets have different dimensions, but are they the same in microns-per-pixel? Do the different sizes of the image affect the performance measures and test time?

4. The model training section is missing. On what images was the model trained? On what images was it tested? How many images were in each set? Only the DRIVE dataset includes a mention of training and testing sets. Was the same model used on all three datasets?

5. The performance measures used are unclear. Are the performance measures calculated on a per-pixel basis? How was the binary threshold chosen to calculated accuracy, sensitivity, and specificity? There are other measures which are better suited to the segmentation problem, such Dice score. Critically, it is not clear which images the performance metrics were calculated on and whether those images were also used for training.

6. Are the results shown significant? The paper would benefit from showing the results of a significance test of the performance measures across all images using the authors' method and the existing methods. The performance benefit of the authors' method would be more clear if the small improvements (on the other of 0.0035 in AUC and 0.0045 in accuracy) were shown to be significant.

7. The bolding of rows in Table 1, 2, and 3 is inappropriate as it implies that the bolded row had the best performance in every performance measure, which is not the case.

**Special Issue:**

No

---

### Review · AnonReviewer1 · 2018-05-07
**Review of Multi-scale Neural Networks for Retinal Blood Vessels Segmentation**

**Rating:** 2
**Confidence:** 2

**Review:**

This paper proposes a supervised blood vessel segmentation technique for retinal images. Their architecture has two main parts: using the features extracted in different scales (using the Atrous Convolutions with different rates); and aggregating the features extracted at low and high levels (similar to the idea of U-Net architecture [1] ). Even though their proposed architectures have novelties and the results reported in their work are good, but there are several strong drawbacks in this article which are mentioned in the following.
1- There are several typos and grammatical mistakes in this article. The poor English has highly decreased the quality of the paper.
2- The main reason why the blood vessel segmentations are needed in clinical practice has not been explained clearly. The main signs of several diseases such as diabetic retinopathy are not found in the blood vessels, but they are detected from the background of the retinal images (e.g. the microaneurysms, hemorrhages, etc.).
3- There are several incorrect claims in the introduction section and the authors have not supported their claims with enough reasons. As an example:
a) Unsupervised approaches doesn’t require labeled data, which can be considered as an advantage. However, the performance of such approaches is far below the satisfaction in real application scenarios and most of them are time consuming. ⇒ this is not a correct argument. The recent unsupervised segmentation techniques are both fast and accurate enough (See [2]).
b) One of these weaknesses is the insufficient use of low-level features which is helpful in semantic segmentation.⇒ This is wrong claim and it is not supported with any proofs, reasons or references.
c) Another weakness is that features extracted by existing approaches are not effective.⇒ This claim has not been supported with any reasons.
4- The references referred to, in Section 1 and 2, or the ones used for the performance comparisons are not the most recent references (See [2]).
5- There is no explanation why the multi-scale features (Fig. 1-2) have been extracted only from the input image in the first level and not the other levels. Moreover, Fig. 1-2 is a bit confusing as the blocks of different scales are connected to each other, while they are independent based on their explanation.
6- The authors have not explained their motivation for using an upsampling layer in initial layers of their network.
7- In Section 4.2 (Augmentation) the authors have explained that they crop 64x64 patches as an augmentation step. So it implies that the inputs of their networks are not the full-sized retinal images but smaller patches. This is in contrast to Fig. 1-1.
8- In Section 4.3 (Structure prediction) the definitions of the sp-1, sp-32, sp-64 are not clear enough; whether they are the different patch sizes used in the augmentation step (Section 4.2, augmentation) or not.
9- The evaluation metrics explained in Section 4.3 are not the best options for evaluating the performance. There are more appropriate performance criteria for evaluating the vessel segmentation performance such as the F1-score or the Matthews correlation coefficient (MCC) as also used in [4].
10- By qualitative comparison between the segmentation results and the ground truth images shown in Fig. 4, it is clear that even though a multi-scale approach has been used, the method is still not capable of segmenting the very tiny vessels very well. The tiny vessels contain clinically important information and are crucial to be detected accurately.
11- In Section 5.4, the authors have mentioned that the Sp-1 achieves the test time of 3s, which is 7s faster than the state-of-the-art unsupervised method. There are more recent works proposed in the literature with better performances than the ones used in this paper for comparison. As an example, the running time reported by [2] as an unsupervised segmentation technique is only 4s, which is as fast as the results reported in this work.
[1] Ronneberger, O. et al. (2015, October). U-net: Convolutional networks for biomedical image segmentation. In International Conference on Medical Image Computing and Computer-Assisted Intervention (pp. 234-241). Springer, Cham.
[2] Zhang, J. et al. (2016). Robust Retinal Vessel Segmentation via Locally Adaptive Derivative Frames in Orientation Scores. IEEE Transactions on Medical Imaging. 35. 1-1. 10.1109/TMI.2016.2587062.
[3] Ricci et al. (2007). Retinal blood vessel segmentation using line operators and support vector classification,” IEEE Transactions on medical imaging, vol. 26, no. 10, pp. 1357–1365.
[4] Azzopardi, G. et al. (2015). Trainable COSFIRE filters for vessel delineation with application to retinal images,  Medical Image Analysis, vol. 19, pp. 46–57.

**Special Issue:**

No

---

### Review · AnonReviewer3 · 2018-05-17

**Rating:** 3
**Confidence:** 1

**Review:**

A Multi-scale NN is designed for segmentation of blood vessels in retinal images. The multi-scale is obtained by so-called atrous convolution. That is a convolution with an extra parameter r that controls what we normally call the stride when subsampling images. I’m really in doubt of this method isn’t equivalent to making a multi-channel cnn with subsampled patches? It is just a stretching of the filter instead of a subsampling of the image? Results are excellent.

**Special Issue:**

No

---

### Comment · ~Zhiming_Luo1 · 2018-04-16
**How much did the preprocessing improve the performance?**

1. As mentioned in the paper, an ACE step is used for preprocessing, and how much did the step improve the performance?
2. From the paper, i don't understand what is  the  sp-1, sp-32 and sp-64?

---

> ### Comment · ~Boheng_Zhang1 · 2018-04-16
> **preprocessing**
>
> Hi Zhiming,
>
>
> 1. The performance improves 0.2%+ in STARE dataset, and other datasets also have some improvments, but less than in STARE.
>
> 2. Sp is the using of sliding window in test. The windows of sp-1, sp-32 and sp-64 are the whole image, 32 × 32 and 64 × 64 respectively. The stride of sp-1, sp-32 and sp-64 are 1, 3 and 3 respectively.
>
> Hope it is helpful.
>
> Boheng

---

### Comment · ~Xiaoyang_Chen1 · 2018-04-18
**How much did your architecture outperform a simple U-Net?**

1. Why you first upsample the input image? Any improvement of this operation compared with the architecture without it?
2. The multi-scale architecture you used is confusing. All the operations are done on the input image? How many feature maps is produced after each operation? And it seems redundant using so many different operations and concatenate them together. Any loss of performance caused by removing any one or more operations?
3. The two convolutional architecture is essentially the idea of Residual block but you did not cite their paper.

---

> ### Comment · ~Boheng_Zhang1 · 2018-04-18
> **the results of u-net**
>
> Hi, Xiaoyang,
>
>
> The results of u-net architecture have been shown in the Tables.
>
> 1、We think it can add features to prevent the disadvantages of conv (stride 2) and other operation. It helps in our single test and we will do more test on the upsample in the future.
>
> 2、All feature maps can be calculated according to the conv parameters shown in the Figures. (U can download the datasets and calculate all the feature map according to the different size of images in different datasets). Our architecture is not perfect, and we may remove or add operations for a new version in the future.
>
> 3、We use a idea of concatenate in lots of convolutional architecture and the using of batch normalization and relu is common these years. It is different from the residual block (for example, the concatenate location) after I see the paper of residual network. Of course, it is better to cite the paper. Thank you.
>
>
> Hope it is helpful.
>
> Boheng

---

### Decision · Program_Chairs · 2018-05-15
**Paper68 Acceptance Decision**

Reject